# Effects of Environmental Conditions on Cathode Degradation of Polymer Electrolyte Fuel Cell during Potential Cycle †

**Yoshiyuki Hashimasa \*, Hiroshi Daitoku and Tomoaki Numata**

E-mobility Research Division, Japan Automobile Research Institute 2530 Karima, Tsukuba, Ibaraki 305-0822, Japan; dhiroshi@jari.or.jp (H.D.); tnuma@jari.or.jp (T.N.)

\* Correspondence: yhashi@jari.or.jp

† Presented at EVTeC and APE Japan on October 2, 2018.

**Abstract:** We investigated the effects of cell temperature and the humidity of gas supplied to the cell during the load cycle durability test protocol recommended by The Fuel Cell Commercialization Conference of Japan (FCCJ). Changes in the electrochemically active surface area (ECA) and in the amount of carbon support corrosion were examined by using the JARI standard single cell. The ECA declined more quickly when the gas humidity was raised, and the carbon corrosion was at the same level. These results suggest that the agglomeration of platinum was accelerated by the same agglomeration mechanism, i.e., by raising the humidity of the gas supplied to the cell.

**Keywords:** polymer electrolyte fuel cell; membrane electrode assembly; platinum degradation; potential cycle

---

## 1. Introduction

To facilitate the widespread use of fuel cell vehicles, it is important to improve the durability and reduce the cost of fuel cell stacks. As a method of testing the durability of membrane electrode assemblies (MEAs), potential cycle tests that simulate the operation of a fuel cell vehicle are widely used. In particular, the load cycle durability test, which simulates acceleration and deceleration under automotive use, is used as an accelerated durability test of the platinum catalyst. It takes a long time to conduct an accelerated durability test. In order to evaluate materials with further improved durability from now on, it is effective to further shorten the test time without changing the deterioration phenomenon. In this study, the influence of environmental conditions during the load cycle durability test on the agglomeration rate of platinum was investigated. The influence of environmental conditions such as temperature and humidity on the decline of ECA during potential cycle tests has been investigated. It is reported that the decrease in normalized ECA ratio increases when temperature and humidity are raised [1–4]. It is said that the increase in Pt particle size and the agglomeration are promoted by Ostwald ripening on carbon support and dissolution-re-precipitation through the ionomer phase. It is suggested that low relative humidity operation during load cycling is important in suppressing the Pt degradation from Pt dissolution and Pt particle growth in the cathode catalyst layer [2]. However, the influence of humidity on the agglomeration of Pt is mostly evaluated up to 100% relative humidity, and there are few evaluation examples at supersaturation condition. Furthermore, the effects of environmental conditions on the mechanism of ECA decline are not well understood because such studies did not necessarily evaluate changes in the carbon corrosion rate of the catalyst support and power generation performance. As ECA decreases even if the carbon support corrodes, it is necessary to simultaneously discuss carbon corrosion at the test temperature. Even if

the reduction of ECA is equal, the I-V performance may be different. In this study, we investigated the effects of environmental conditions on the decline of ECA and carbon corrosion rate, taking into consideration the acceleration factor.

## 2. Experimental Method

### 2.1. MEA and Single Cell Specifications

A Pt/C catalyst (TEC10E50E, Tanaka Kikinzoku Kogyo, Tokyo, Japan) and a polymer electrolyte dispersion solution (Nafion® DE2020, DuPont) were mixed to prepare a catalyst paste. The weight ratio between carbon support and ionomer was set at 1:1. The catalyst paste was applied to and dried on a Teflon sheet by the doctor blade method. The cut Teflon sheets sandwiched a Nafion211 electrolyte membrane, with the catalyst layer side of both sheets in contact with the Nafion211 membrane. The sheet–membrane unit was hot-pressed at 135 °C for 10 min. The Pt loading was set at 0.3 mg cm$^{-2}$ for both anode and cathode. An MEA thus prepared and a 28BC gas diffusion layer made by SGL were assembled into a JARI standard single cell [5].

### 2.2. Load Cycle Durability Test

A potential cycle test using a rectangular wave of 0.6–1.0 V vs. RHE (Figure 1) under the environmental conditions shown in Table 1 was carried out as the load cycle durability test.

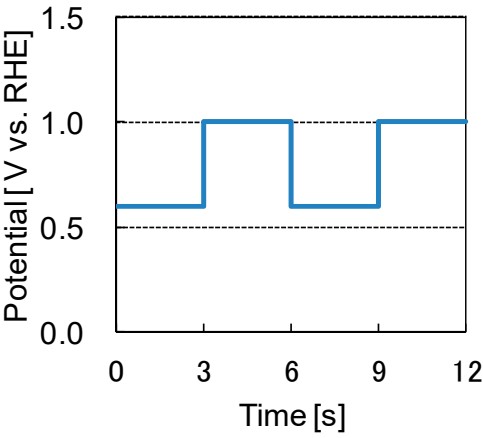

**Figure 1.** Potential waveform of the load cycle test.

**Table 1.** Environmental conditions during load cycle durability test.

| Parameter | Measurement Condition |
|---|---|
| Cell temperature | 70, 80, 90 °C |
| Relative humidity | 65, 100, 150% RH |
| Gaseous species | Hydrogen (anode), Nitrogen (cathode) |
| Flow rate | 0.2 (anode), 0.8 (cathode) L min$^{-1}$ |
| Pressure | Atmospheric |

A cyclic voltammetry measurement for electrochemical surface area (ECA) and a I-V measurement for power generation performance were conducted as diagnostics of the degradation status. The test conditions for cyclic voltammetry (CV) and I-V measurement are shown in Tables 2 and 3, respectively. During the durability test, $CO_2$ concentration in the nitrogen gas exhausted from the cathode was measured with a non-dispersive infrared analyzer in order to determine the progression of carbon support corrosion.

**Table 2.** Conditions of cyclic voltammetry measurement.

| | |
|---|---|
| **Cell Temperature** | 80 °C |
| **Hydrogen Flow Rate** | 0.2 L min$^{-1}$ |
| **Nitrogen Flow Rate** | 0 L min$^{-1}$ |
| **Anode/Cathode Relative Humidity** | 100% |
| **Scan Range** | 0.05–0.9 V vs. RHE |
| **Scan Rate** | 50 mV s$^{-1}$ |
| **Scan Cycle** | 5 |

**Table 3.** Conditions of I-V performance measurement.

| | |
|---|---|
| **Cell Temperature** | 80 °C |
| **Hydrogen Utilization Ratio** | 70% |
| **Air Utilization Ratio** | 40% |
| **Hydrogen Dew Point** | 77 °C |
| **Air Dew Point** | 60 °C |
| **Outlet Pressure** | Atmospheric |

## 3. Results and Discussion

### 3.1. Effects of Cell Temperature

After a predetermined number of cycles, a cyclic voltammetry measurement for electrochemical surface area (ECA) was conducted to determine the degradation status. Figure 2 shows the effect of the cell temperature on the relationship between the normalized ECA ratio and the number of potential cycles in the load cycle durability test. The influence of cell temperature on the rate of ECA decrease was not so large, and was increased a little when the cell temperature was raised.

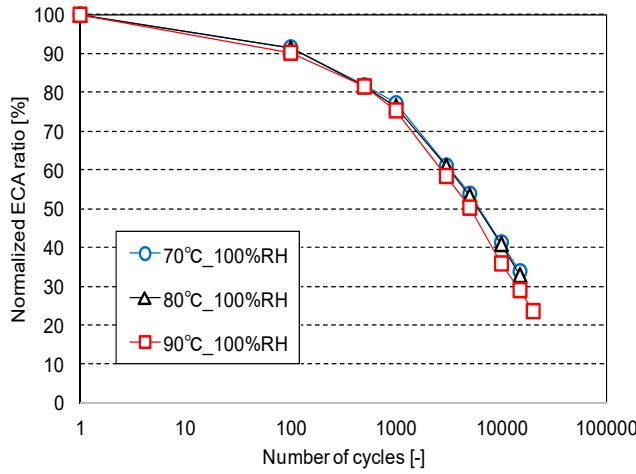

**Figure 2.** Effect of cell temperature on the relationship between the normalized ECA ratio and the number of potential cycles.

In order to calculate the amount of carbon support corrosion during the load cycle durability test, the concentration of $CO_2$ in the nitrogen gas exhausted from the cathode outlet was measured with a non-dispersive infrared analyzer. Figure 3 shows the effect of the cell temperature on the relationship between the normalized carbon corrosion ratio and the number of potential cycles in the load cycle durability test. The influence of cell temperature on the rate of carbon decrease was not so large, and

was increased a little when the cell temperature was raised. From Figures 2 and 3, both ECA and carbon ratio decreased by number of potential cycles, but the decline rate of ECA was larger than that of carbon. The ECA decrease rate in the load cycle test is mainly influenced by the decrease of the Pt surface area due to increased particle size rather than the carbon support corrosion [4–6]. The influence of cell temperature on the Pt agglomeration rate was considered not to be large under the test conditions in this study.

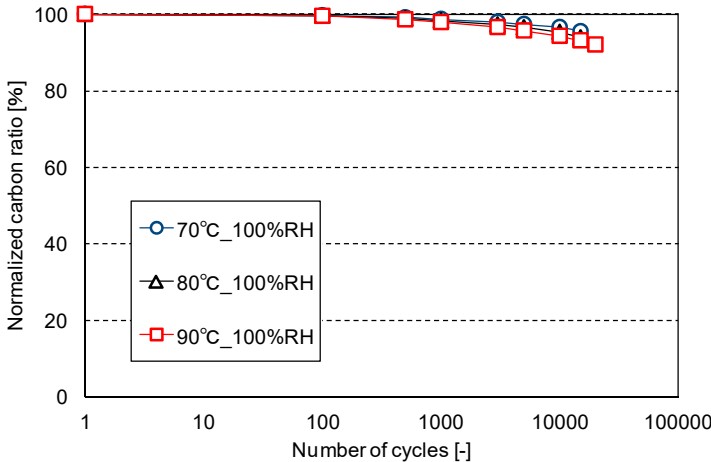

**Figure 3.** Effect of cell temperature on the relationship between the normalized carbon support corrosion ratio and the number of potential cycles.

The durability of the Pt/C catalyst is dependent on the durability of both the carbon support and platinum, and both the degradation of platinum and carbon corrosion decrease the ECA of the Pt/C catalyst. Figure 4 shows the relationship between the normalized ECA ratio and the normalized carbon support corrosion ratio under each cell temperature condition. The relationship was similar under each cell temperature condition, as shown in Figure 4. Therefore, it was thought that an almost identical Pt/C degradation phenomenon occurred even if the cell temperature was different in the load cycle durability test.

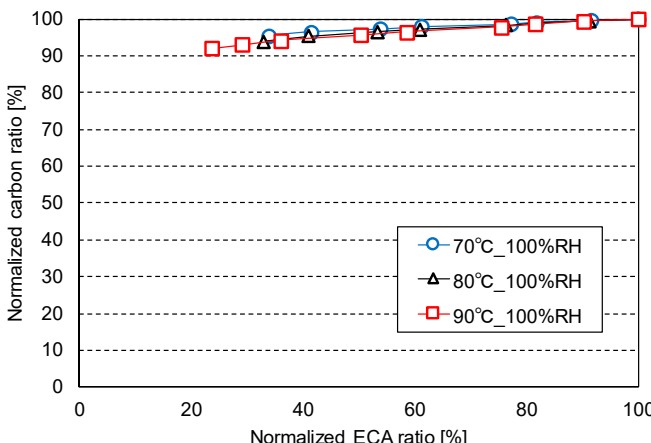

**Figure 4.** Effect of cell temperature on the relationship between the normalized ECA ratio and the normalized carbon support corrosion ratio.

The influence of cell temperature on the decline in power generation performance in the load cycle durability test was investigated by I-V measurements as diagnostics of the degradation status. Figure 5 shows the relationship between the cell voltage at 1 A cm$^{-2}$ and the ECA in the load cycle durability test. No influence of cell temperature on the relationship was observed in the range of these

examination conditions. From the viewpoint of the power generation performance, we confirmed that the same agglomeration phenomenon occurred even if the cell temperature was different in the load cycle durability test.

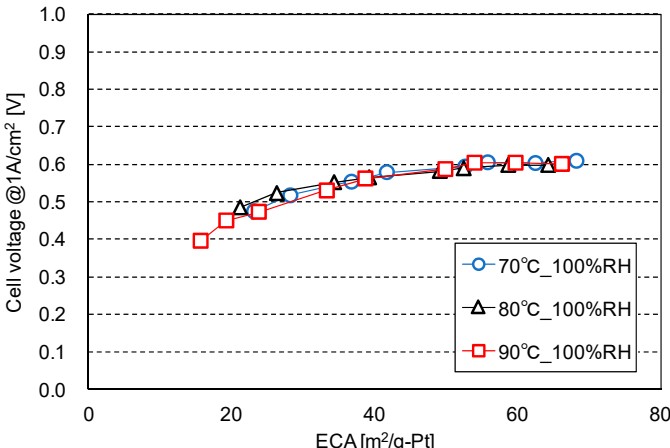

**Figure 5.** Effect of cell temperature on the relationship between the cell voltage at 1 A cm$^{-2}$ and the ECA in the load cycle durability test.

## 3.2. Effects of Humidity

Next, we investigated the influence of the humidity of the gas supplied to the cell on the ECA and carbon corrosion in the load cycle durability test. Figure 6 shows the effect of the humidity of the gas supplied to the cell on the relationship between the normalized ECA ratio and the number of potential cycles in the load cycle durability test. Relative humidity influenced the rate of ECA decrease, which was increased when the humidity was raised.

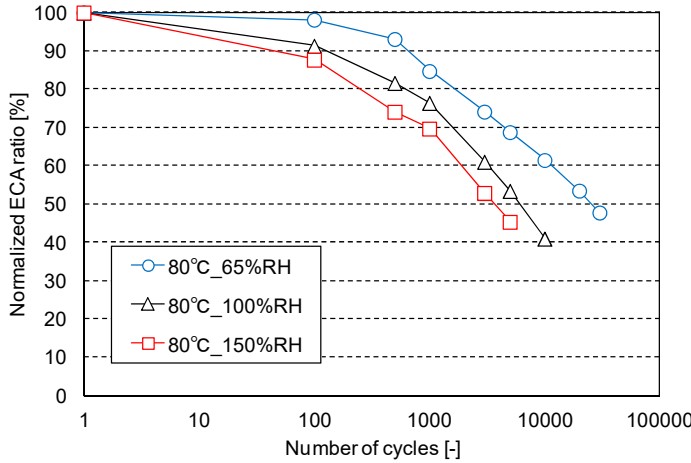

**Figure 6.** Effect of humidity on the relationship between the normalized ECA ratio and the number of potential cycles.

In order to calculate the amount of carbon support corrosion during the load cycle durability test, the concentration of $CO_2$ in the nitrogen gas exhausted from the cathode outlet was measured with a non-dispersive infrared analyzer. Figure 7 shows the effect of the humidity of the gas supplied to the cell on the relationship between the normalized carbon corrosion ratio and the number of potential cycles in the load cycle durability test. The carbon corrosion ratio was not influenced by the relative humidity under the test conditions in this study. It was thought that the higher the humidity, the faster the Pt agglomeration rate. The Pt dissolution rate is thought to be increased due to the large amount of water, but more detailed investigation is necessary.

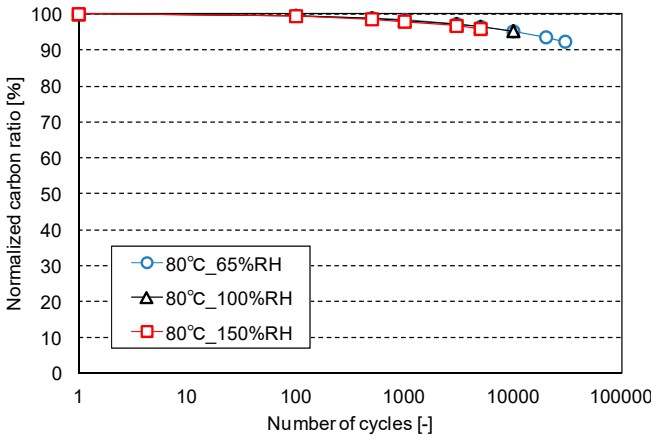

**Figure 7.** Effect of humidity on the relationship between the normalized carbon support corrosion ratio and the number of potential cycles.

Figure 8 shows the relationship between the normalized ECA ratio and the normalized carbon support corrosion ratio under each humidity condition. The relationship was similar under each humidity condition, as shown in Figure 8. Therefore, it was thought that an almost identical Pt/C degradation phenomenon occurred even if the humidity supplied to the cell was different in the load cycle durability test.

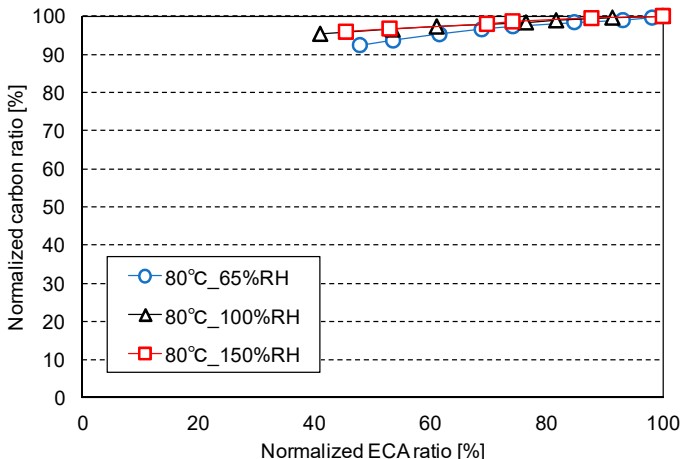

**Figure 8.** Effect of humidity on the relationship between the normalized ECA ratio and the normalized carbon support corrosion ratio.

The influence of humidity supplied to the cell on the decline in power generation performance in the load cycle durability test was investigated by I-V measurements as diagnostics of the degradation status. Figure 9 shows the relationship between the cell voltage at 1 A cm$^{-2}$ and the ECA in the load cycle durability test. No influence of humidity on the relationship was observed in the range of these examination conditions. From the viewpoint of the power generation performance, we confirmed that the same agglomeration phenomenon occurred even if the humidity was different in the load cycle durability test.

Figure 10 shows the effect of humidity on the relationship between the normalized cell voltage at a current density of 1 A cm$^{-2}$ and the number of potential cycles. Relative humidity influenced the rate of cell voltage decrease, which was increased when the humidity was raised. The reason why the rate of cell voltage decrease increased in a smaller number of potential cycles was thought to be that the rate of ECA decrease was higher under more humid conditions.

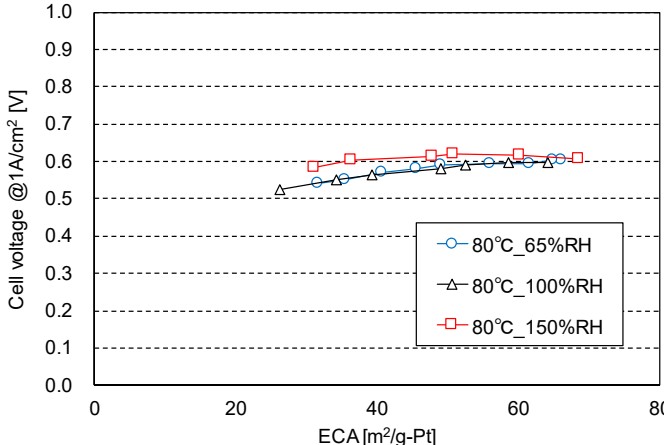

**Figure 9.** Effect of humidity on the relationship between the cell voltage at 1 A cm$^{-2}$ and the ECA in the load cycle durability test.

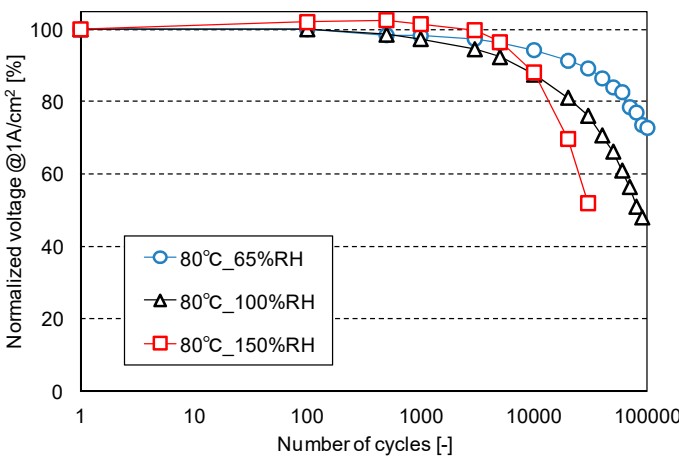

**Figure 10.** Effect of humidity on the relationship between the cell voltage at 1 A cm$^{-2}$ and the number of potential cycles.

In order to investigate in detail the difference in performance degradation for each humidity condition, the amount of performance degradation was divided by each overvoltage [7]. During the power generation test, the cell resistance was measured with an AC resistance meter (10 kHz). The product of the resistance and the current was used as the resistance overvoltage. From the measured cell voltage, the IR-free cell voltage was calculated by the correction of the resistance polarization. The activation overvoltage was then defined as the difference between the theoretical cell voltage and the corresponding voltages of the linear extrapolation of the Tafel region (low current density region) of the IR-free polarization curve. The diffusion overvoltage was the voltage obtained by subtracting the activation overvoltage and the resistance overvoltage from the cell voltage.

Figure 11 shows the changes in the relationship between overvoltage and ECA in the load cycle durability test. ECA decreases according to the number of potential cycles. A remarkable increase was not seen in resistance overvoltage or diffusion overvoltage when ECA decreased. Regardless of the humidity, it was considered that there was no deterioration in the performance of the electrolyte membrane or ionomer due to the potential cycle test and no change in the pore structure of the catalyst layer. The activation overvoltage increased with a decline of ECA. It was believed that the agglomeration of Pt particles reduced the reaction area required for power generation. No influence of humidity was seen in the increase in activated overvoltage. These results show that the performance degradation mechanisms caused by the load cycle test were almost the same regardless of humidity conditions.

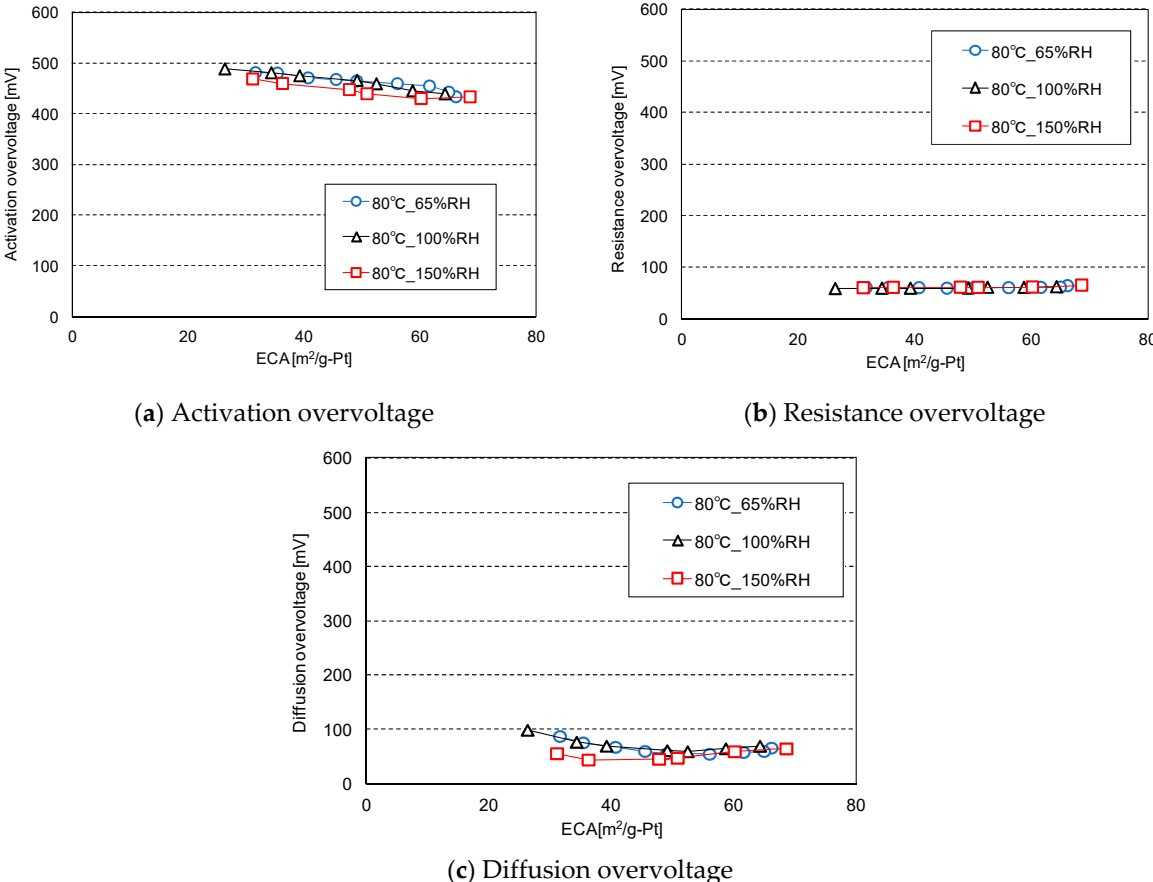

(**a**) Activation overvoltage  (**b**) Resistance overvoltage

(**c**) Diffusion overvoltage

**Figure 11.** Effect of humidity on the relationship between overvoltage and ECA at a current density of 1 A cm$^{-2}$ during the load cycle test.

Therefore, it was confirmed that the decline of ECA with the number of potential cycles was accelerated at high humidity.

## 4. Conclusions

We investigated the effects of cell temperature and the humidity of gas supplied to the cell on the stability of the platinum catalyst during the load cycle durability test, in order to shorten the testing time without changing the deterioration phenomenon of the carbon-supported platinum catalyst. The effects of environmental conditions on the decline of ECA, carbon corrosion rate and cell performance were examined simultaneously. It was revealed that humidity influenced the cell performance more than temperature did, because the influence of humidity on the speed of ECA decrease was larger than that of the cell temperature. Also, the carbon corrosion ratio was not influenced by the relative humidity under the test conditions in this study. The Pt dissolution rate is thought to be increased by raising humidity due to the large amount of water. It was also revealed that the mechanism of the Pt agglomeration was thought to be the same when the humidity of the gas was changed, by analyzing the platinum catalyst degradation and its effect on the power generation performance in detail. It may be possible to reduce the testing time for load cycle durability by raising the humidity of the gas supplied to the cell.

**Author Contributions:** Conceptualization, Y.H.; Data curation, H.D. and T.N.

**Funding:** This research was funded by New Energy and Industrial Technology Development Organization, grant number 15100824-0.

**Conflicts of Interest:** The authors declare no conflict of interest.

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
