# Peer review of "Effects of Environmental Conditions on Cathode Degradation of Polymer Electrolyte Fuel Cell during Potential Cycle†"

_wevj, doi:10.3390/wevj10020024_

Reviewer 1 Report

The manuscript describes the impact of humidity and operating temperature on the proton exchange membrane fuel cell durability. The topic is important for the commercialization of fuel cell vehicles. However, I don't know which part of the study presented in this manuscript is new to the community. This topic has been studied extensively in the last 10 years or so, and we can easily find many published studies. Furthermore, the voltage loss after voltage cycling is presented but the reasons for such voltage loss are not discussed. Therefore, I don't think this manuscript is good for journal publication unless the originality of this study is clearly presented and the results are further analyzed.

Author Response

The manuscript describes the impact of humidity and operating temperature on the proton exchange membrane fuel cell durability. The topic is important for the commercialization of fuel cell vehicles. However, I don't know which part of the study presented in this manuscript is new to the community. This topic has been studied extensively in the last 10 years or so, and we can easily find many published studies. Furthermore, the voltage loss after voltage cycling is presented but the reasons for such voltage loss are not discussed. Therefore, I don't think this manuscript is good for journal publication unless the originality of this study is clearly presented and the results are further analyzed.

I thank you for your peer review and accurate comments. We highlighted the motivation and background of this study in the introduction. As for the voltage loss after voltage cycling, we added discussion about the reasons for such voltage loss.

Reviewer 2 Report

The manuscript addressed the effect of gas humidity and temperature on Pt agglomeration and carbon support corrosion in catalyst layers during potential cycles applied to PEFC. For better clearance of delievering the purpose, some points of view should be revised for publication as follows:

1. The word, Pt deterioration, was not appropriate. Please substitute it into dissolution or agglomeration describing the phenomena occurring in catalyst layers during potential cycles. 

2. Many reports were made to decribe the effect of environmental conditions on cathode degradation of PEFC including humidity and temperature. The authors should highligt the motivation and background of this study in the Introduction section. (e.g., refer to Introduction in Energies, 2019, 12(3), 549)

3. At Line 69, The amount of decrease in the~ should be polished. Please revise the related expression throughout the manuscript.

4. Please unify the color of the symbols indicating 65, 100 and 150% in Figure 6-11.

5. At Line 160-161, the sentence, 'It was revealed that the
161 influence of humidity on the speed of ECA decrease was larger than that of the cell temperature,' is not appropriate. It should mean that humidity influenced the performance greater than temperature.

6. In this manuscript, at the same humidity, operation temperature did not influence ECA, carbon support oxydation, and performance during the potential cycles. However, humidity greatly influence ECA change and performance. The authors mentioned somewhat simple conclusion based on the aforementioned observation. It seems that the main purpose of this manuscript investigate the effect of temperature and humidity on Pt agglomeration and carbon support oxidation (or loss) during the potential cycles.  ECA could be influenced by a decrease of Pt surface area due to Ostwald ripening, carbon support loss or swelling of ionomer binder in catalyst layers. Since the authors did not provide TEM image which could describe the Pt size distribution before and after potential cycle, it is harldy concluded that Pt agglomeration affected a decline in ECA. The authors confirmed that there are no effect of humidity and temperature on carbon support loss by the measurement of CO2 coming out of cathode. Figure 6, 10 and 11 might conclude that the swelling of ionomer binder could be larger at higher humidity to decrease ECA and porosity of catalyst layers. The authors should explain much clearer why humidity makes performance and activation/diffusion overpotential much worse than temperature in the revised manuscript. 

Author Response

Please find the response in the attached file. 

Round  2

Reviewer 1 Report

I think this revised version is ready to publish.

Author Response

I thank you for your peer review and accurate comments.

We fixed a spelling error in the introduction line 35. (supersaturration→supersaturation)

We fixed a spelling error in the conclusion line 187. (humity→humidity)

We added the explanation int the conclusion line 184 and 189.

We fixed the format of reference lists.
